# Xarifiid Copepods (Copepoda: Cyclopoida: Xarifiidae) Parasitic in the Coral *Psammocora columna* Dana, 1846 from Taiwan

**DOI:** 10.3390/ani11102847

**Published:** 2021-09-29

**Authors:** Yu-Rong Cheng, Tsai-Ming Lu, De-Sing Ding

**Affiliations:** 1Department of Fisheries Production and Management, National Kaohsiung University of Science and Technology, No. 142, Haijhuan Rd., Nanzih District, Kaohsiung 811213, Taiwan; 2Institute of Cellular and Organismic Biology, Academia Sinica, No. 128, Academia Rd., Section 2, Nankang District, Taipei 115024, Taiwan; tmlu@gate.sinica.edu.tw; 3Ph.D. Program of Aquatic Science and Technology in Industry, College of Hydrosphere Science, National Kaohsiung University of Science and Technology, No. 142, Haijhuan Rd., Nanzih District, Kaohsiung 811213, Taiwan; 1041750102@nkust.edu.tw

**Keywords:** coral-associated copepods, new species, parasitism, scleractinian corals, taxonomy

## Abstract

**Simple Summary:**

The coral reef is the crucial habitat for numerous marine creatures, and scleractinian corals are building blocks of this community. Therefore, the health condition of scleractinian corals is an essential factor for the sustainability of coral reef ecosystems. Recent studies indicate that global warming, seawater acidification as well as coral diseases are the main threats to scleractinian corals. In addition, coral endoparasites may impact the health of scleractinian corals by consuming coral tissues and potentially transferring disease pathogens. However, we have limited knowledge about the distributions of coral endoparasites across scleractinian corals. It may lead to the assessment bias on the health condition of the coral reef community due to no consideration on the impact of the interaction between corals and their endoparasites. Here, we performed an elaborate survey on a widely distributed scleractinian coral species, *Psammocora columna*, and discovered two new parasitic copepod species, *Xarifia*
*yanliaoensis* and *Xarifia magnifica*. We also summarized a classification key of morphological characteristics for the identification of *Xarifia* copepods in *Psammocora* corals. The findings of this study present new records of copepod-coral relationships in the Indo-West Pacific for the biological resource database. Furthermore, these are the footstone knowledge for further studies on coral reef conservation.

**Abstract:**

A comprehensive knowledge of relationships between coral and coral-associated organisms is essential for the conservation studies of the coral reef community, yet the biodiversity database of coral-inhabiting copepods remains incomplete. Here we surveyed in a widely distributed scleractinian coral, *Psammocora columna* Dana, 1846, and newly discovered two endoparasitic copepod species, *Xarifia*
*yanliaoensis* sp. nov. and *Xarifia magnifica* sp. nov. These two new species are described based on specimens collected in Taiwan, and they share several common morphological characters of *Xarifia* copepods, i.e., region dorsal to fifth legs having three posteriorly directed processes unequally. However, *X*. *yanliaoensis* sp. nov. is distinguishable from other species by the morphology of the endopods of legs, antenna, maxilla, and maxilliped (in both genders). The morphological characters of *X*. *magnifica* sp. nov. are the endopods of legs, leg 5, and maxilliped in the male. Including the two new species described in the present work, the genus *Xarifia* Humes, 1960 belongs to the cyclopoid family Xarifiidae Humes, 1960 currently consists of 94 species, and eight of them live in association with the *Psammocora* coral. A comparison table and a key to the species of *Xarifia* from *Psammocora* corals are given herein.

## 1. Introduction

Parasitic copepods use a wide range of scleractinian corals as hosts [1]. Up to date, a total of 363 copepod species representing 99 genera, 19 families and three orders (Cyclopoda, Siphonostomatoida, and Harpactioida) have been recognized as parasites in 148 shallow-water stony corals [2]. The total included 288 cyclopoids, 68 siphonostomatoids, and seven harpacticoids. These coral-associated copepods exhibit a marked diversity in morphology in accordance with their respective ecological niches. Among the coral-associated copepods, Xarifiidae Humes, 1960 [3] is a family of endoparasitic copepods living in the gastrovascular cavities of coral polyps and more than 90 valid species have been discovered in Indo-West Pacific coral reefs [2,4,5]. It has been evident that their virulence may be related to their life history strategies, *Symbiodinium* density, surface area of host coral colonies, and concentration of nitrate and chlorophyll-*a* in the surrounding seawater. Therefore, the potential of using these parasites as bioindicators for predicting the future physiological performance of host corals in response to environmental change can be developed by tracking their abundance and species composition [2].

According to the previous studies carried out from 1967 to 2010 [4,6,7,8,9,10], 15 species of parasitic copepods have been discovered from a widely distributed scleractinian genus, *Psammocora* Dana, 1846 [11] throughout the Indo-West Pacific, including three highly modified cyclopoids: *Xarifia diminuta* Humes and Ho, 1967 [6], *Xarifia formosa* Humes, 1985 [4], *Xarifia imitans* Humes, 1985 [4]. Recently, Cheng and Lin reported three new species of *Xarifia*, *Xarifia conrepta* Cheng and Lin 2021 [12], *Xarifia gracilis* Cheng and Lin 2021 [12], and *Xarifia lata* Cheng and Lin 2021 [12], parasitic in *Psammocora digitata* Milne Edwards and Haime 1851 [13] from Taiwan. Thus, among 92 *Xarifia* species recorded up to date, six of them have been discovered in *Psammocora* corals. All six species can be found in *P*. *digitata*, while *X. diminuta* also occurred in *Psammocora contigua* [14]. (Table 1).

Obviously, the current knowledge about coral-associated copepods of the widespread *Psammocora* corals remains limited. There are ten valid species of *Psammocora* around the world [15], but only two species of *Psammocora* corals have been examined for coral-associated copepods. *Psammocora columna* Dana, 1846 [11] is a widely distributed coral and has not been examined for coral-associated copepods yet. Herein, we perform a survey in two colonies of *P*. *columna* collected from a shallow-water reef on the north coast of Taiwan (Figure 1), and describe two new species of xarifiid copepods. Combining our findings with previous records, 20 species of parasitic copepods including eight species of *Xarifia* have been found in *Psammocora* corals.

## 2. Materials and Methods

### Specimen Collection

Several fragments of two colonies of *Psammocora columna* Dana, 1846 [11] were sampled by scuba diving on coral reefs in northern Taiwan (Figure 1 and Figure 2). Coral fragments were placed in plastic bags underwater and transported to the laboratory. We adopted the standard methods from previous studies to collect copepod specimens [4,6,12]. Briefly, coral samples in sea water were placed in a 500 mL beaker and enough 95% ethyl alcohol was added to make an approximately 5% solution, which was left to sit for at least 4–6 h. Then, coral fragments were removed and the liquid with any sediment was poured through a fine net (approximately 100 µm mesh size). The copepods then were acquired from the sediment using forceps and preserved in 70% ethanol. Individuals were later cleared in 85% lactic acid for 1–2 h, then dissected on a wooden slide under a dissecting microscope [16]. The appendages were examined under a compound microscope using magnifications of up to 1000×. All drawings were made with the aid of a drawing tube.

## 3. Results

SYSTEMATICS. Family Xarifiidae Humes, 1960. Genus *Xarifia* Humes, 1960. *Xarifia*
*yanliaoensis* sp. nov. (Figure 3, Figure 4 and Figure 5).

Material examined: Three specimens (two ♀♀ and one ♂) were obtained from washings of a *Psammocora columna* colony collected at 5 m depth, at Longdon (25°06′49.5″ N, 121°55′11.8″ E), Gongliao District, New Taipei, Taiwan on 12 August 2010 (Figure 1 and Figure 2). The other specimens (two ♀♀ and three ♂♂) were obtained from washings the other colony of the same coral species collected at 9 m depth, at Yanliao (25°03′04.4″ N 121°56′01.7″ E) on 20 August 2010 (Figure 1 and Figure 2). Female holotype (NTUM-Inv-10019), male allotype (NTUM-Inv-10020); paratypes (NKUST-Cop-0004) were deposited in National Taiwan University Museum (NTUM), Taipei, Taiwan.

Female: Body (Figure 3A,B) slender, about 6.16 times longer than wide. Length 1.17 (1.07–1.26) mm and greatest width 0.19 (0.17–0.21) mm based on two specimens in lactic acid before dissection. External segmentation not evident, but prosome with 4 lateral constrictions. Three unequal long posteriorly directed processes on the region dorsal to fifth legs, the middle one slightly smaller (Figure 3A–C). Genital somite and abdomen (Figure 3C) 3-segmented, straight or slightly curved dorsally, together occupying ~25% of body. Genital openings dorsolateral (Figure 3A–C). Caudal ramus (Figure 3D) elongate, covered with small setules, 107 µm long, 31 µm wide at the base, ratio 3.45:1, bearing 6 setae (1 outer lateral and 5 terminal setae). Body surface smooth (Figure 3A–C).

Antennule (Figure 3E) 5-segmented, width of first to fifth segments: 23.3 × 18.7, 18.7 × 15.6, 12.5 × 10.9, 7.8 × 7.8, and 9.4 × 6.2 μm, respectively; armature formula: 3, 13, 5 + 1 aesthetasc, 2 + 1 aesthetasc, and 7 + 1 aesthetasc from proximal to distal; all setae naked. Antenna (Figure 4A) 4-segmented. Armature formula 1, 1, 2, 2+claw from proximal to distal; terminal claw slightly shorter than final segment. Mandible (Figure 4B) simple, slender, and smooth. Maxillule (Figure 4C) with 2 terminal setae and 1 small anterior spiniform process. Maxilla (Figure 4D) 2-segmented; first segment (syncoxa) unarmed; second segment (basis) drawn out into pointed process having large, slightly curved, claw-like seta with lamella, and bearing with two unequal inner setae. Maxilliped (Figure 4E,F) 3-segmented; syncoxa (first segment) with large, distal protuberance (the length as the second segment of maxilliped); basis (second segment) having lateral lobe, 2 medial setae proximally; endopod (third segment) small, tipped with 2 spiniform setae.

Legs 1–4 (Figure 5A–D) each with 3-segmented exopod, 2-segmented endopod. The shape and armature of leg 2 as in leg 1, but endopod of leg 2 armed with 2 setae (instead of 1 seta). Exopods of legs 3 and 4 similar to legs 1 and 2, but 2 setae only (instead of 3 setae) on the terminal segment. Endopod of leg 3 unarmed. Endopod of leg 4 bearing with 1 seta. Formula of spines (Roman numerals) and setae (Arabic numerals) as follows:

**Coxa****Basis****Exopod****Endopod**Leg 10-01-0I-0; I-0; I, 30-0; 1Leg 20-01-0I-0; I-0; I, 30-0; 2Leg 30-01-0I-0; I-0; I, 20-0; 0Leg 40-01-0I-0; I-0; I, 20-0; 1

Leg 5 (Figure 3A–C) 155.4 μm long, 44.4 μm wide at base, tapering distally, armed with single proximal seta and 2 apical setae.

Male: Body (Figure 6A,B) slender than female, about 8.6 times longer than wide. Length 1.38 (1.23–1.52) mm and greatest width 0.16 (0.15–0.16) mm based on two specimens in lactic acid before dissection. Caudal ramus (Figure 6C) small, about 25.0 × 19.1 μm, as in female, bearing 5 setae. Body surface smooth.

The appendages including antennule, antenna, mandible, maxillule, and maxilla resembling that of female, but antennule with additional aesthetasc indicated by a dot on second segment (Figure 3E). Maxilliped (Figure 6D) 4-segmented; syncoxa (first segment) and first endopodal segment (third segment) unarmed; basis (second segment) with 2 medial setae; second endopodal segment (fourth segment) a claw with bifurcate tip, 2 proximal setae at base; concave margin with 3 serrations.

Legs 1–4 resembling those of female; leg 5 (Figure 6B) minute, with 2 terminal setae, dorsal seta adjacent to the base. Leg 6 (Figure 6B) as 2 small setae on genital operculum of genital somite.

Spermatophore not observed.

Etymology: The species is named after the type-locality Yanliao.

*Xarifia magnifica* sp. nov. (Figure 7, Figure 8 and Figure 9).

Material examined: Four specimens (two ♀♀ and two ♂♂) were obtained from washings of a *Psammocora columna* colony collected at 5 m depth, at Longdon (25°06′49.5″ N, 121°55′11.8″ E), Gongliao District, New Taipei, Taiwan on 12 August 2010 (Figure 1 and Figure 2). Female holotype (NTUM-Inv-10021) and male allotype (NTUM-Inv-10022) were deposited in National Taiwan University Museum (NTUM), Taipei, Taiwan.

Female: Body (Figure 7A,B) moderately stout. Length1.45 (1.33–1.58) mm and greatest width 0.27 (0.25–0.29) mm based on two specimens in lactic acid before dissection. Ration of body length to greatest width 5.4:1. External segmentation distinct, prosome having 4 lateral constrictions. Region dorsal to fifth legs with 3 equal long posteriorly processes (Figure 7A–C). Genital somite and abdomen (Figure 7C) 3-segmented, straight, together occupying ~20% of body length. Genital openings dorsolateral (Figure 7A–C). Egg sacs not observed. Caudal ramus (Figure 7D) elongate, 107.4 × 27.9 µm, bearing with 4 or 6 setae. Body surface smooth (Figure 7A–C).

Antennule (Figure 7E) 4-segmented, armature: 3, 18 + 1 aesthetasc, 2 + 1 aesthetasc, 4 + 1 aesthetasc; all setae naked. Antenna (Figure 8A) 4-segmented, armature: 1, 1, 2, 2+claw; terminal claw small, about 1/2 length of terminal (fourth) segment. Mandible (Figure 8B) with smooth blade. Maxillule (Figure 8C) with 3 setae distally, including one small anterior spiniform process and two relatively long middle setae. Maxilla (Figure 8D) 2-segmented; similar to that of *X. yanliaoensis* sp. nov., but second segment (basis) slightly small and bearing with relatively longer outer seta. Maxilliped (Figure 8E,F) 3-segmented; syncoxa (first segment) with a small, distal protuberance; basis (second segment) with 2 lateral lobes, 2 medial setae proximally; endopod (third segment) much small, bearing with 2 elements (the larger one equal to the length of third segment).

Legs 1–4 (Figure 9A–D) each with 3-segmented exopod, 2-segmented endopod. The shape and armature of leg 2 as in leg 1. Exopods of legs 3 and 4 similar to legs 1 and 2, but their third segment armed with 2 setae and 1 seta, respectively. Endopod of legs 3 and 4 unarmed. Formula of spines (Roman numerals) and setae (Arabic numerals) as follows:

**Cox**a**Basis****Exopo**d**Endopod**Leg 10-01-0I-0; I-0; I, 30-0; 2Leg 20-01-0I-0; I-0; I, 30-0; 2Leg 30-01-0I-0; I-0; I, 20-0; 0Leg 40-01-0I-0; I-0; I, 10-0; 0

Leg 5 (Figure 7A–C) as that in *X. yanliaoensis* sp. nov., but slightly stout, about 275.5 μm long and 75.5 μm wide at base.

Male: Body (Figure 10A,B) slender than female, about 8.2 times longer than wide. Length 1.55 (1.53–1.57) mm and greatest width 0.19 (0.16–0.21) mm based on two specimens in lactic acid before dissection. Abdomen (Figure 10A) segmentation indistinct. Caudal ramus (Figure 10C) small, about 55.8 × 27.9 μm, ornamented with several setules, bearing with 4 terminal setae (lateral seta not clearly evident). Body surface covered with several setules (Figure 10A,B).

The appendages including antennule, antenna, mandible, maxillule, and maxilla as that of female, but antennule with additional aesthetasc indicated by a dot on second segment (Figure 7E). Maxilliped (Figure 10D) 4-segmented; syncoxa (first segment) and first endopodal segment (third segment) unarmed; basis (second segment) largest, with 2 medial setae (one normal and one modified with sclerotized base and hyaline attenuate distal part); second endopodal segment (fourth segment) a claw with bifurcate tip, bearing with 2 proximal setae equally; concave margin with 3 serrations.

Legs 1–4 as in female; leg 5 (Figure 10B) consisting only of 3 setae. Leg 6 (Figure 10A,B) represented by 2 small setae on the genital operculum of genital somite.

Spermatophore not observed.

Etymology: The specific name *magnifica*, Latin adjective for “big” or “majestic” refers to the relatively strong element on the third segment of maxilliped.

## 4. Discussion

Based on the findings from this study, the genus *Xarifia* currently consists of 94 species and eight of them live in association with the *Psammocora* coral. The armature of endopods of legs 1–4 is the main diagnostic feature of *Xarifia*. The terminal segment of endopods of legs 1–4 of *X*. *yanliaoensis* sp. nov. is a unique character, armed with 1, 2, 0, 1 setae, respectively. After comparison with another valid species of *Xarifia*, we recognized that only *Xarifia simplex* Humes, 1985 [4] parasitic in the *Scapophyllia cylindrica* (=*Merulina cylindrica* (Milne Edwards and Haime, 1849) [17]) present the same armature of endopods of legs 1–4. However, the body surface of *X*. *simplex* covered with scattered small hairs (setules), while the body surface of *X*. *yanliaoensis* sp. nov. is smooth. The *X*. *yanliaoensis* sp. nov. differs from *X*. *simplex* by (1) the armature formula of the maxilla: three setae in *X*. *yanliaoensis* sp. nov. but two in *X*. *simplex* [4] (p. 561: Figure 49l); (2) the armature of terminal segment of antenna: I + 2 in *X*. *yanliaoensis* sp. nov., while I + 1 in *X*. *simplex* [4] (p. 561: Figure 49h); (3) the size of lobe or protuberance on the second segment of maxilliped: the protuberance in *X*. *yanliaoensis* sp. nov. is obviously bigger than that in *X*. *simplex* [4] (p. 562: Figure 50a,b) and its length is similar to the second segment of maxilliped. In addition, the maxilliped of male between these two species also showed some differences: with trifurcate tip and triangular process on the concave edge of the fourth segment in *X*. *simplex* [4] (p. 562: Figure 50j), but with bifurcate tip and 3 serrations on concave margin in *X*. *yanliaoensis* sp. nov.

The other new species described herein, *X*. *magnifica* sp. nov., is similar to *Xarifia anopla* Humes and Dojiri, 1982 [18], *Xarifia brevicauda* Humes and Ho, 1968 [19], *Xarifia filata* Humes, 1985 [4], *Xarifia hadra* Humes and Dojiri, 1983 [20], *Xarifia scutipes* Humes and Dojiri, 1983 [20], *Xarifia longa* Cheng, Ho and Dai, 2007 [21], *Xarifia capillata* Cheng, Ho and Dai, 2011 [22], and *Xarifia parva* Cheng, Ho and Dai, 2011 [22] in the armature formula of the terminal endopodal segments of legs 1–4 (2, 2, 0, 0). *Xarifia anopla*, *X*. *filata*, *X*. *hadra*, *X*. *longa*, and *X*. *parva* can be excluded first because of their endopods of legs 1-4 exhibit only 1 segment. *Xarifia capillata* can be distinguished from *X*. *magnifica* sp. nov. by the region dorsal to fifth legs with a single central process which is tipped with tuft of setules [22] (p. 228: Figure 1A–C). *Xarifia scutipes* may be distinguished easily from *X*. *magnifica* sp. nov. in the shield-like leg 5 in the female [20] (p. 282: Figure 18a–c) and the claw of the maxilliped in the male with a large hyaline excrescence on the concave margin [20] (p. 285: Figure 21d,e). *Xarifia brevicauda* differs from *X*. *magnifica* sp. nov. by its abbreviated genital somite and abdomen [19] (p. 448: Figures 114–116), and by the serrated excrescence on the claw of the male maxilliped [19] (p. 449: Figure 130).

Eight species of *Xarifia* have been known using *Psammocora* corals as hosts (Table 1). We selected certain external features of *Xarifia* that are useful for the determination of species. Eleven features are shown in Table 2 and a useful key is provided as follows:


**Key to females of eight species of *Xarifia* parasitic in *Psammocora* corals (modified from [12]).**


1a. Region dorsal to fifth legs with processes or knobs…2

1b. Region dorsal to fifth legs smooth… *X. lata*

2a. Sides of genital segment smooth…3

2b. Sides of genital segment with small sclerotized lobe…*X**. imitans*

3a. Maxillule with 3 elements…4

3a. Maxillule with 2 elements…6

4a. Maxilliped with 2 lobes…5

4b. Maxilliped with 3 lobes…*X*. *magnifica* sp. nov.

5a. Terminal armature on endopods of legs 1 (4 2, 3, 0, 1)…*X**. conrepta*

5b. Terminal armature on endopods of legs 1 (4 1, 2, 0, 1)…*X*. *yanliaoensis* sp. nov.

6a. Maxilliped with more than 2 lobes…7

6b. Maxilliped with 2 lobes…*X. diminuta*

7a. Terminal armature on endopods of legs 1 (4 2, 3, 0, 1)…*X. formosa*

7b. Terminal armature on endopods of legs 1 (4 3, 3, 2, 2)…*X**. gracilis*

**Table 2 animals-11-02847-t002:** Differences between the eight species of *Xarifia* copepods isolated from *Psammocora* corals.

(a)						
Species	Characters *					
	1	2	3	4	5	6
*X*. *diminuta*	0.98 (0.78–118)	5.4:1	3	4	2, 2, 0, 2	3, 3, 2, 2
*X*. *formosa*	1.15 (1.10–1.19)	5.2:1	3	5	2, 3, 0, 1	3, 3, 3, 2
*X*. *imitans*	0.97 (0.94–1.01)	6.6:1	3	5	2, 3, 0, 1	3, 3, 3, 2
*X*. *conrepta*	0.80 (0.74–0.82)	4.7:1	3	5	2, 3, 0, 1	3, 3, 2, 2
*X. gracilis*	0.78 (0.75–0.80)	6.5:1	3	5 or 6	3, 3, 2, 2	3, 3, 2, 2
*X*. *lata*	1.05 (1.02–1.09)	5.8:1	0	4	1, 2, 1, 1	2, 2, 2, 1
*X. magnifica* sp. nov.	1.45 (1.33–1.58)	5.4:1	3	4 or 6	2, 2, 0, 0	3, 3, 2, 1
*X.**yanliaoensis* sp. nov.	1.17 (1.07–1.26)	6.2:1	3	6	1, 2, 0, 1	3, 3, 2, 2
**(b)**						
**Species**	**Characters ***					
	**7**	**8**	**9**		**10**	**11**
*X*. *diminuta*	2	2	2 setae		2 lobes	smooth
*X*. *formosa*	2	2	2 setae		4 lobes	smooth
*X*. *imitans*	2	2	2 setae		2 lobes	with lateral process
*X*. *conrepta*	2	2	3 setae		2 lobes	smooth
*X. gracilis*	2	2	2 setae		5 lobes	smooth
*X*. *lata*	1 or 2	1 or 2	2 setae		3 lobes	smooth
*X.magnifica* sp. nov.	2	2	3 setae		3 lobes	smooth
*X.**yanliaoensis* sp. nov.	2	2	3 setae		2 lobes	smooth

* The characters are based on females. Leg armature is expressed in terms of spines (in Roman numerals) and setae (in Arabic numerals). Characters 1: body size (mm); 2: ratio of length to width; 3: No. of processes above fifth legs; 4: No. of setae on caudal ramus; 5: terminal armature on endopods of legs 1–4; 6: terminal armature on exopods of legs 1–4. Characters 7: No. of segments on endopods of legs 1–2; 8: No. of segments on endopods of legs 3–4; 9: armature of maxillule; 10: armature of maxilliped; 11: genital segment.

We believe that the abundance and species composition of coral-associated copepods in the host corals could be considered as bioindicators for predicting the health condition of host corals. Furthermore, this could further be an assessment index for coral conservation. However, this assessment could be feasible only if the interactions and ecological impacts between parasitic copepods and host corals are well recognized. In the present study, we have improved the understanding of species diversity of coral-associated copepods in the world. As accumulating knowledge of parasitic copepod and host coral relationships, we expect to reduce the stress of harms and diseases to host corals by applying biological control of parasitic copepods in the future.

## 5. Conclusions

To fulfil distribution information of xarifiid copepods in reef-building corals and enrich the completeness of the biological resource database of the Indo-West Pacific area, we studied *Xarifia* copepods in Taiwan for decades. So far, 22 species of *Xarifia* have been described as endoparasites of corals in Taiwanese waters. Although *P. columna* is a widely distributed coral, there is no available information on the parasite fauna of this coral species. In the present study, we examined several fragments of two colonies of this coral species for parasitic copepods and described two new species of *Xarifia*. Totally, 20 species of parasitic copepods including eight species of *Xarifia* are now known to be associated with *Psammocora* corals. Furthermore, we summarized a classification key of eight *Xarifia* species in *Psammocora* corals, which would be a handy index for a broader of coral reef researchers to identify copepod species by themselves. The discovery of so many species of parasitic copepods from *Psammocora* corals is unusual. Further studies to investigate whether the distinct morphological traits of these parasitic copepods might increase the fitness and allow them to adapt to their micro-niches inside the corals are required. Uncovering the phylogenetic relationships and interactions between these parasitic copepods and corals may enrich the fundamental knowledge to conservation biologists for assessments of health conditions of coral reef community, as well as the co-evolution between host–parasite relationships.

## Figures and Tables

**Figure 1 animals-11-02847-f001:**
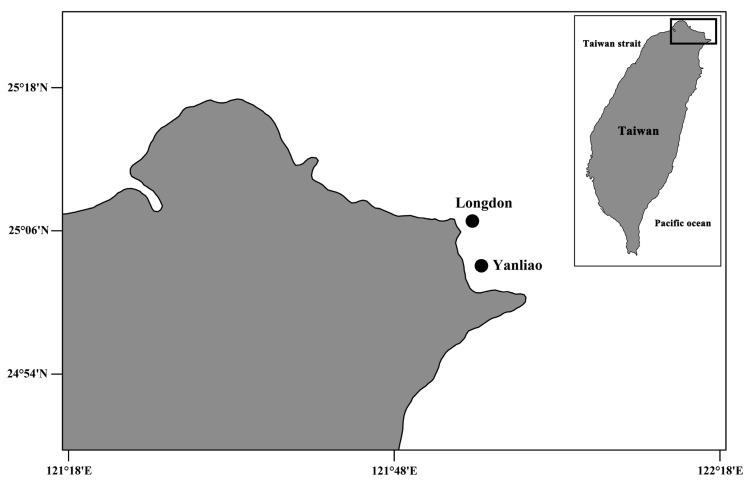
Map of Taiwan showing the type locality of *Xarifia*
*yanliaoensis* sp. nov. and *Xarifia magnifica* sp. nov.

**Figure 2 animals-11-02847-f002:**
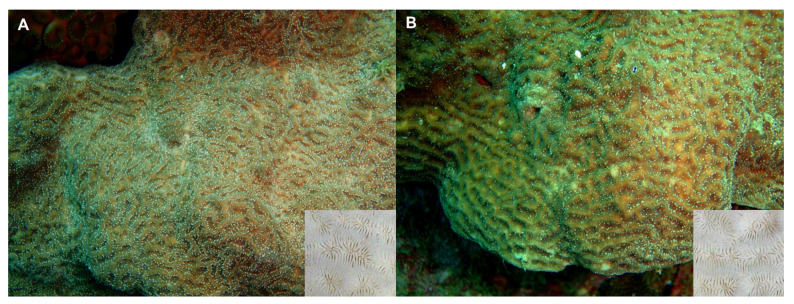
Photographs of two colonies of *Psammocora columna* Dana, 1846 collected by scuba diving on coral reefs in northern Taiwan and examined for coral-associated copepods in this study. (**A**) A colony of *P*. *columna* collected at 5 m depth at Longdon; (**B**) a colony of *P*. *columna* obtained at 9 m depth at Yanliao. The inserted photos show skeleton structure of each coral colony.

**Figure 3 animals-11-02847-f003:**
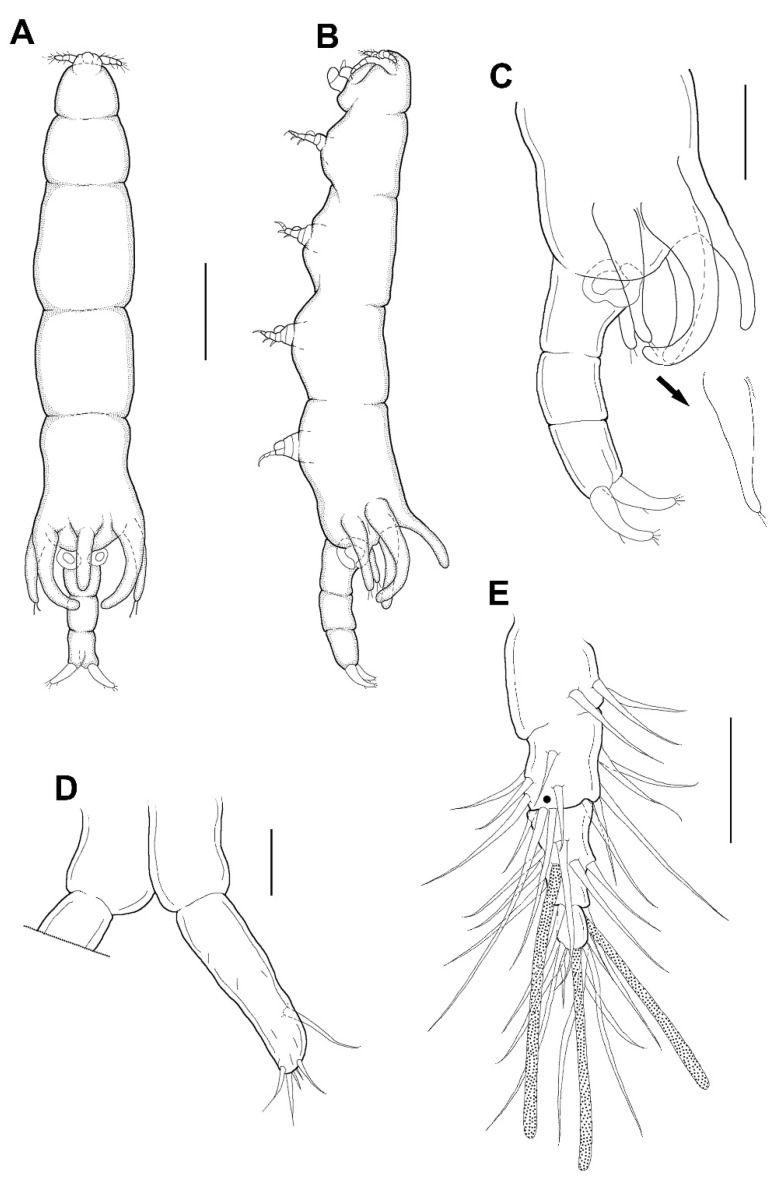
*Xarifia**yanliaoensis* sp. nov. female. Habitus, dorsal (**A**); habitus, lateral (**B**); urosome, lateral; arrow indicate leg 5 (**C**); caudal ramus (**D**); antennule (**E**). Scale bars: (**A**,**B**) = 0.2 mm, (**C**)= 0.1 mm; (**D**,**E**) = 0.025 mm.

**Figure 4 animals-11-02847-f004:**
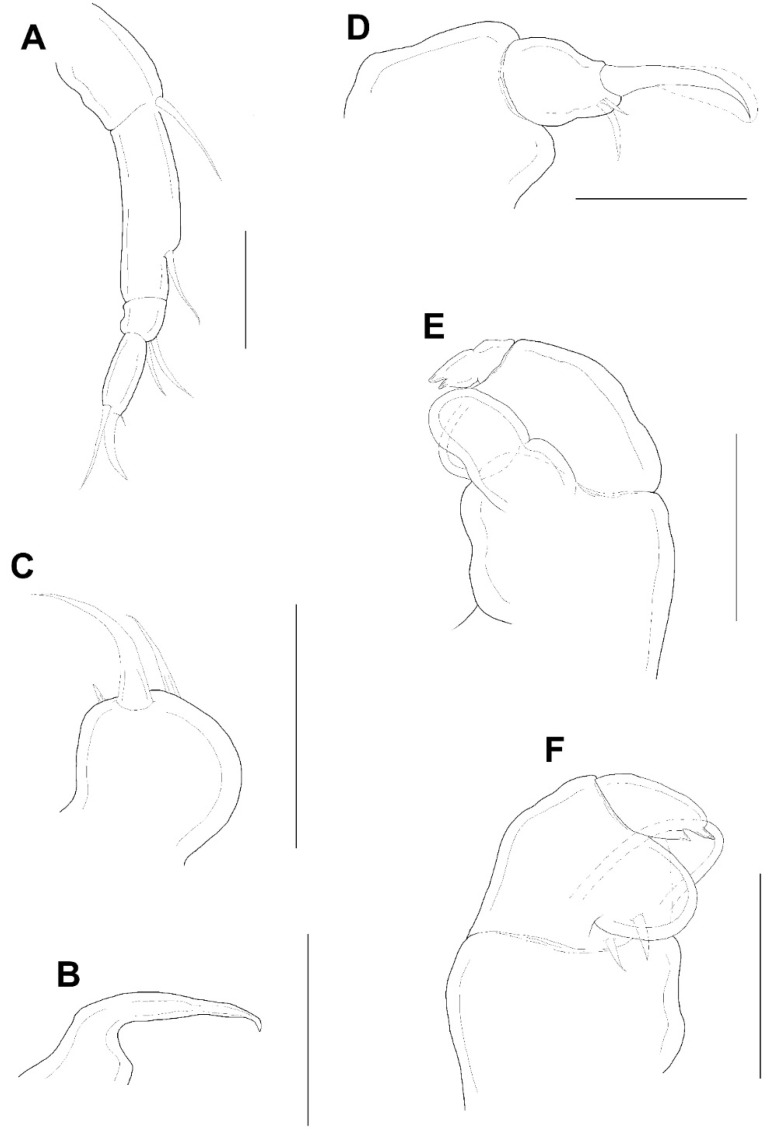
*Xarifia**yanliaoensis* sp. nov. female. antenna (**A**); mandible (**B**); maxillule (**C**); maxilla (**D**); maxilliped (**E**); other face of maxilliped (**F**). Scale bars: (**A**–**F**) = 0.025 mm.

**Figure 5 animals-11-02847-f005:**
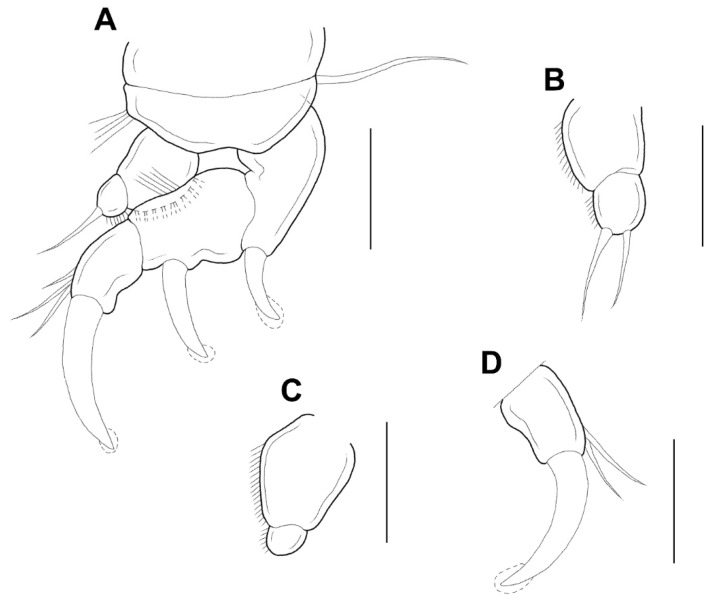
*Xarifia**yanliaoensis* sp. nov. female. leg 1 (**A**); endopod of leg 2 (**B**); endopod of leg 3 (**C**); terminal segment of exopod of leg 3 (**D**). Scale bars: (**A**–**D**) = 0.025 mm.

**Figure 6 animals-11-02847-f006:**
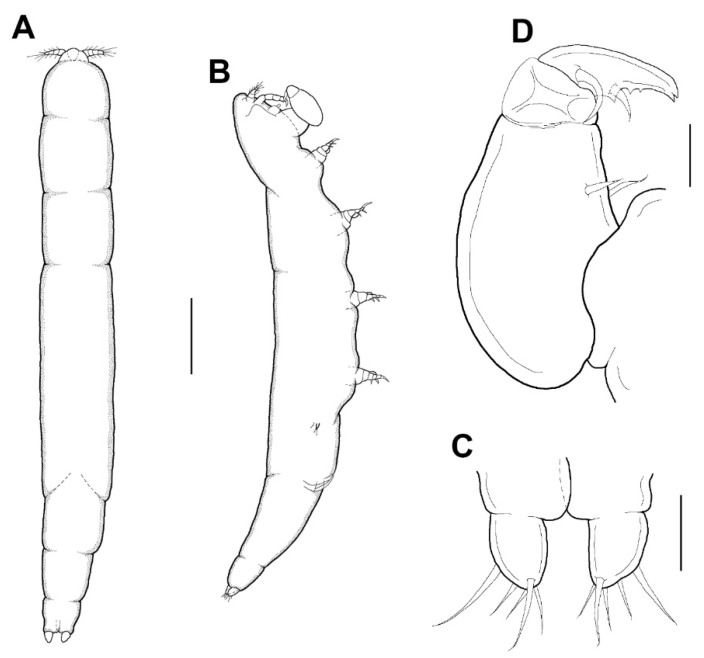
*Xarifia**yanliaoensis* sp. nov. male. Habitus, dorsal (**A**); habitus, lateral (**B**); caudal ramus (**C**); maxilliped (**D**). Scale bars: (**A**,**B**) = 0.2 mm; (**C**,**D**) = 0.025 mm.

**Figure 7 animals-11-02847-f007:**
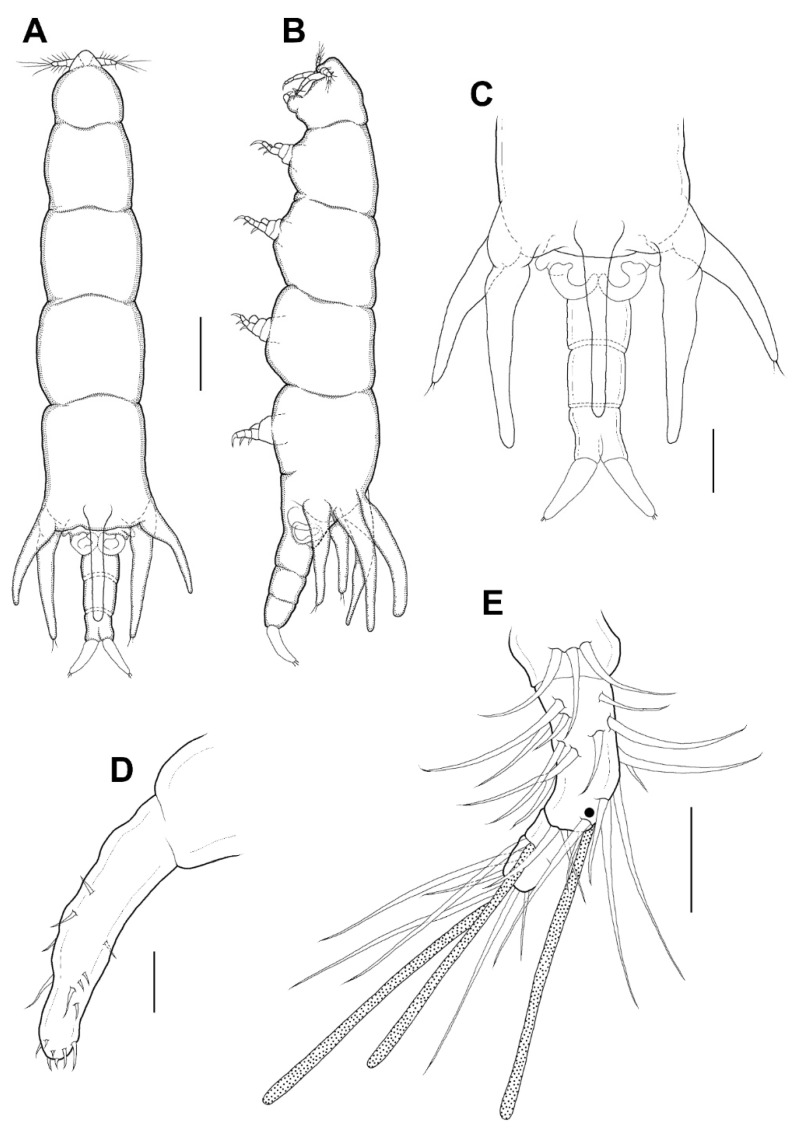
*Xarifia magnifica* sp. nov. female. Habitus, dorsal (**A**); habitus, lateral (**B**); urosome, dorsal (**C**); caudal ramus (**D**); antennule (**E**). Scale bars: (**A**,**B**) = 0.2 mm, (**C**) = 0.1 mm; (**D**,**E**) = 0.025 mm.

**Figure 8 animals-11-02847-f008:**
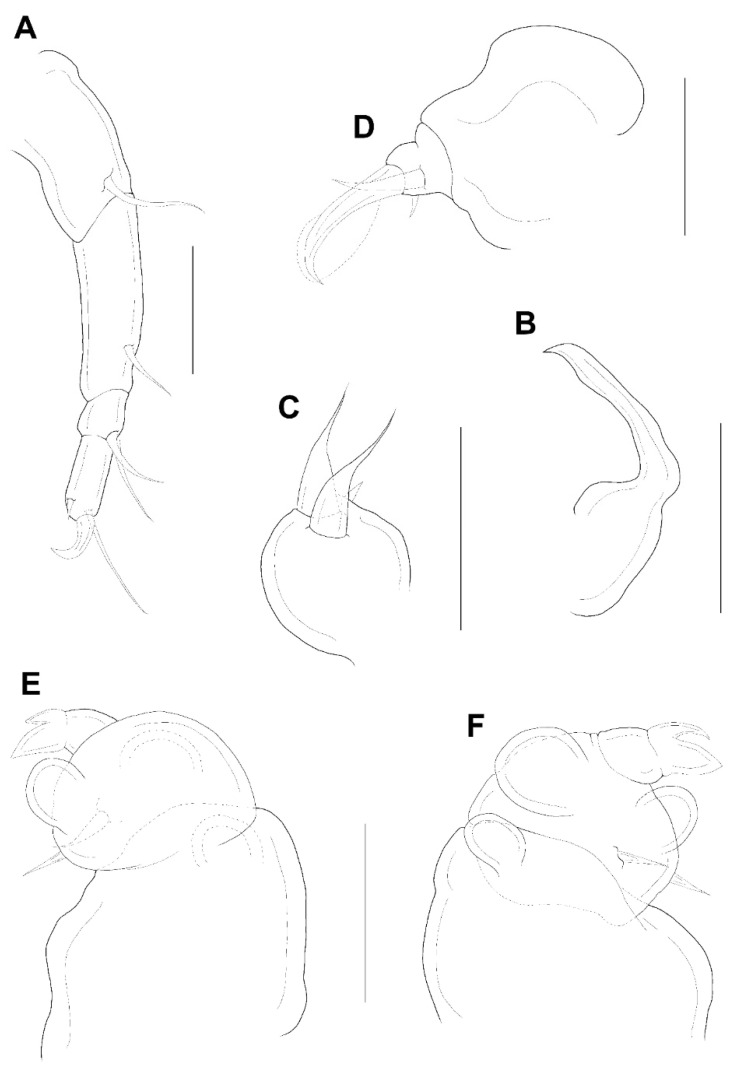
*Xarifia magnifica* sp. nov. female. antenna (**A**); mandible (**B**); maxillule (**C**); maxilla (**D**); maxilliped (**E**); other face of maxilliped (**F**). Scale bars: (**A**–**F**) = 0.025 mm.

**Figure 9 animals-11-02847-f009:**
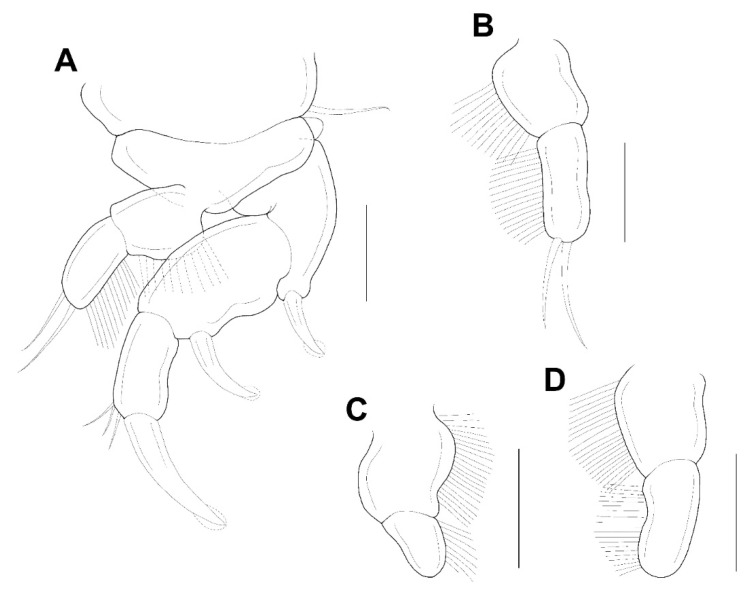
*Xarifia magnifica* sp. nov. female. Leg 1 (**A**); endopod of leg 2 (**B**); endopod of leg 3 (**C**); endopod of leg 4 (**D**). Scale bars: (**A**–**D**) = 0.025 mm.

**Figure 10 animals-11-02847-f010:**
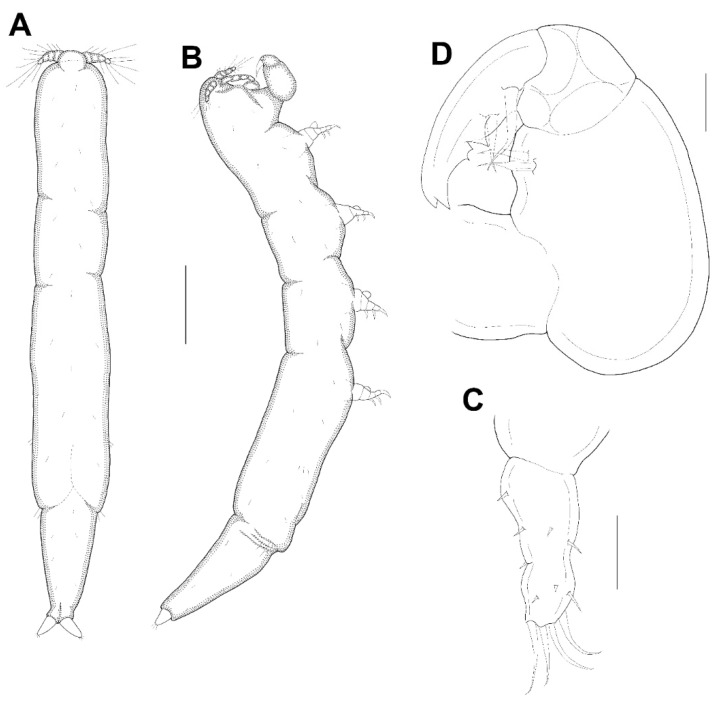
*Xarifia magnifica* sp. nov. male. Habitus, dorsal (**A**); habitus, lateral (**B**); caudal ramus (**C**); maxilliped (**D**). Scale bars: (**A**,**B**) = 0.25 mm; (**C**,**D**) = 0.025 mm.

**Table 1 animals-11-02847-t001:** Eight species of *Xarifia* copepods associated with *Psammocora* corals.

Copepod Parasite	Host Coral	Distribution	Reference
*Xarifia diminuta* Humes and Ho, 1967	*Psammocora contigua* (Esper, 1794)	Madagascar	[6]
	*Psammocora* cf. *contigua*	New Caledonia	[4]
	*Psammocora digitata* Milne Edwards and Haime, 1851	Taiwan	[12]
*Xarifia formosa* Humes, 1985	*P*. *digitata*	New Caledonia	[4]
*Xarifia imitans* Humes, 1985	*P*. *digitata*	New Caledonia	[4]
	*P*. *digitata*	Marshall Islands	[4]
	*P*. *digitata*	New Caledonia	[4]
*Xarifia conrepta* Cheng and Lin, 2021	*P*. *digitata*	Taiwan	[12]
*Xarifia gracilis* Cheng and Lin, 2021	*P*. *digitata*	Taiwan	[12]
*Xarifia lata* Cheng and Lin, 2021	*P*. *digitata*	Taiwan	[12]
*Xarifia yanliaoensis* sp. nov.	*Psammocora columna* Dana, 1846	Taiwan	herein
*Xarifia magnifica* sp. nov.	*P*. *columna*	Taiwan	herein

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
