# Peer review of "Xarifiid Copepods (Copepoda: Cyclopoida: Xarifiidae) Parasitic in the Coral *Psammocora columna* Dana, 1846 from Taiwan"

_animals, 2021, doi:10.3390/ani11102847_

Round 1

Reviewer 1 Report

A very good article describing two new species of Xarifia copepods parasitic in scleractinian corals. Also, a comparison table and a key to the species of Xarifia from 40 Psammocora corals are given. English language and style are fine. Therefore my recommendation is to accept in present form.

Author Response

A very good article describing two new species of Xarifia copepods parasitic in scleractinian corals. Also, a comparison table and a key to the species of Xarifia from 40 Psammocora corals are given. English language and style are fine. Therefore, my recommendation is to accept in present form.

Reply: We deeply thanks for your approval.

Reviewer 2 Report

This is an interesting reports on two new parasitic copepods from coral host. This can be enough importance for publishing in Animals. I only think two points to tell.

1) Is "simple Summary" essential for the paper? I think it can be get rid of from the manuscript.

2)Page 7, line 146. Authors claim P5 armed with single proximal and 2 apical setae, however, the single proximal seta is not clear from the figure 3A-C. I do suggest that authors can give an enlarged figure for the seta, and make it clearer for showing its presence.

After the minor correction, the paper can be published in this journal.

Reviewer 3 Report

Review for the paper "Xarifiid Copepods (Copepoda: Cyclopoida: Xarifiidae) Parasitic in the Coral Psammocora columna Dana, 1846 from Taiwan" by Yu-Rong Cheng, Tsai-Ming Lu and De-Sing Ding submitted to "Animals".

This work deals with xarifiid copepods, endoparasites in scleractinian corals, from Taiwan. The goal of the study was to study Psammocora columna, a widely distributed coral from a shallow-water reef on the north coast of Taiwan, focusing on two new species of coral-associated copepods. Considered limited data and publication dealing with this topic, I feel that this is a novel paper but in the current form it may be interesting only for a local audience. Island coastal ecosystems play an important ecological role in the marine environment. Coral reefs have great significance by their enhanced productivity, the highest diversity of flora and fauna, the complexity of the trophic structure and the resources that have direct economic importance to humanity. Coral reefs of Taiwan may be considered as one of the world's most spectacular marine ecosystems with extremely diverse environmental variations. Associations with many species may have a negative and positive impact on the host corals with parasites having great significance. The authors collected copepods using standard methods. Sample processing and laboratory treatment were done using standard methods. The morphological description was preformed based on modern zoological studies and guides. In general, the description of two new taxa, systematic, and key to 8 species of the genus Xarifia are well written; there are many Figures and Tables to illustrate the main findings. Discussion Section is rather short and may be significantly improved. After minor revisions, the paper may be recommended for publication in Animals.

General comment.

  1. Introduction. Please, provide more data on the biology of parasitic copepods inhabiting different coral species. Describe, shortly, possible negative impacts of Xarifia on corals from Taiwan (mortality estimations, population losses, damage etc.).

  1. Introduction. Table 1. References must be formatted according to the Journal's rules, Use [1], [2], etc. instead of Humes & Ho (1967), Humes (1985), etc.

  1. 3. Discussion. Please, discuss possible applications of your results for coral conservation, reduction of damages and recovery of coral populations.

  1. Discussion. Please, compare your data with other studies dealing with copepod-coral associations worldwide.

Some text revisions are made in the attached PDF.
